# Is there any difference between the owners and the public in their visual impact assessments?——A case study of the front garden of multi-storey residential buildings

Tao Gu[1], Chenping Han [1]*, Mengmeng Zhao[2]*, Lin Zhang[1], Zhongju Yang[1], Jian Zhang[2,3]

1 School of Architecture and Design, China University of Mining and Technology, Xuzhou, China, 2 School of Design, Shanghai Jiao Tong University, Shanghai, China, 3 China Institute for Urban Governance, Shanghai Jiao Tong University, Shanghai, China

* hanchenping@cumt.edu.cn (CH); 394650544@qq.com (MZ)

## Abstract

As a special garden unique to China, the front garden of multi-storey residential buildings has certain public attributes but is managed by a single owner. In this study, the front gardens of multi-storey residential buildings in Jiangsu province, China, were set as research object. Meanwhile, the size of garden, plant type, plant vertical structure, color number, plant coverage, and fence material were chosen as the landscape features to be explored. Then the experiments were conducted to reveal the visual impact assessment rendered by the public with different demographic attributes and the results obtained were analyzed. As is indicated by the statistical analysis, significant differences exist between the owners and the public in their visual impact assessments of the front gardens; the six landscape features are the main factors that influence the public's visual impact assessment; and the public with different demographic attributes would render different visual impact assessments of front gardens. This study offers valuable help for the design of front gardens of multi-storey residential buildings.

## 1. Introduction

### 1.1 The front garden of multi-storey residential buildings

As the most extensive residential building form, the multi-storey unit living mode is deemed as the most intensive application of land in China [1]. At present, home buyers generally seek high-quality and adequate living conditions [2]. Accordingly, the real estate developers of newly-built communities pay more and more attention to community greening, which has emerged as a key selling feature for real estate development ventures [3]. As a symbol of nature, garden landscapes are a central component of real estate marketing rhetoric [4]. Therefore, a large number of real estate developers put forward the marketing strategy of "buying first-floor residence and getting a front garden for free", which wins the favor of many buyers [5]. However, according to Article 73 of *Property law of the People's Republic of China*, the

**Data Availability Statement:** The minimal data set underlying the results are within the paper and Supporting Information files.

**Funding:** The author(s) received no specific funding for this work.

**Competing interests:** The authors have declared that no competing interests exist.

green lands within the building area, except for the public green lands of cities and towns or those which are definitely ascribed to individuals, shall be commonly owned by all owners. Therefore, the owners of the first-floor residences only have the management right of front gardens, not the property right. Different from the private gardens of private villas and garden houses, the front garden of multi-storey residential buildings is a special garden: it has certain public properties but is managed by a single owner, which is unique to China [6].

Some scholars maintained that the garden attached to residences functions as the green space most accessible to the residents [7]. It plays a crucial role in enhancing the residential environment [8]. When the residence owner possesses a garden, his/her health index would be improved [9, 10]. Suyin Chalmin-Pui, Roe [11] conducted experiments in northern England and found that compared with the bare front yard, the front garden of their house is more satisfactory to residents as a result of their renovation. The residents feel that their health has improved as a result. Larsen and Harlan [12] revealed that residential landscape design plays an important role in the life quality of homeowners, and homeowners with different demographic attributes have different preferences for the landscape design of their residential yards.

Meanwhile, scholars also observed that it is of highly necessity to assess the visual contact effect of the public with green space [3]. Much has been done on the significant difference between landscape designers and the public in their preferences over landscapes [13–15]. However, there is relatively little research on differences between homeowners and the public in their visual impact assessments of the front garden of multi-storey residential buildings. The green spaces near the residence and their types affects the relationship among neighbors, which in turn influences the dwellers' happiness [7]. Therefore, it is of great significance to study the visual preference of the owners and the public for the front garden of multi-storey residential buildings, which will improve the design of residents' living environment.

## 1.2 Visual impact assessment

Visual impact assessment is frequently employed to assess how various architectural elements affect people's visual preferences. Askari [16] studied the public's visual impact assessment of buildings with this method and found that architectural style and colors affect the public's visual impact assessment of building façade. Pan, Yuan [17] studied the exterior form of Chinese court buildings and found that architectural style, aspect ratio, and open space in front of the building are the main factors that affect people's visual impact assessment of court buildings. The way individuals perceive green spaces is rooted in their holistic environmental experiences. With vision serving as the most immediate sensory encounter, this method is often used to evaluate the quality of the environment [3]. Visual impact assessment has been proved to be a mature method to study the environmental aesthetics of residential surroundings [18]. Many scholars have applied this method to their research on people's preferences for different landscape elements in their residential environment. Huang and Sherk [19] discovered that rain garden, vegetable, and parking affect public's visual impact assessment significantly. Turgut, Atabeyoğlu [20] used this method to evaluate the shape, color, texture, and height of plants in Cumhuriyet neighborhood. Zhang, Yuan [21] studied the visual impact assessment of the landscapes in the residential area rendered by residents living at different heights. In addition, many other scholars have argued that architectural resilience also affects the public's visual impact assessments [22–24]. Their findings show that landscape style, green area, color number and vertical structure are the primary factors influencing people's visual impact assessment. Nevertheless, there remains a dearth of research examining the disparities between the owners and the public in their visual impact assessments of front gardens in multi-storey residential buildings.

## 1.3 Demographic attributes

Some scholars claimed that demographic attributes would influence people's perception of visual effects [25, 26]. Iftikhar, Asghar [27] conducted a cross-cultural survey of signage design elements and visual preferences. Their findings show that participants with different cultural backgrounds show significant differences in visual preferences for targets. People with different educational levels would render different visual impact assessments of the exterior form of buildings [28]. Molnarova, Sklenicka [29] discovered that people with different educational levels display significant difference in their visual preferences for wind turbines and landscape combinations. Pan, Yuan [17] found that gender, age, educational level are the main demographic attributes that affect the public's visual impact assessment of Chinese court buildings. Suyin Chalmin-Pui, Roe [11] conducted intervention experiments on front garden in northern England and found that participants' gender, age, race, educational level, income, and employment status all affect the experimental results significantly. Häfner, Zasada [30] conducted a research on the visual impact assessment of agricultural landscapes and found that participants' educational level, gender, and age are highly influential to their visual impact assessments. Other scholars hold that gender [31], age [32, 33], professional knowledge [34, 35], value orientation [36, 37], occupation [38, 39] and income [40, 41] all exert significant influence on people's visual impact assessment.

Based on previous studies and combined with the characteristics of research objects, this study selected four demographic attributes, namely, gender, age, education level, and income. The first three characteristics are the most commonly involved elements in relevant studies on landscape assessment [42–45], while the last one has been verified to be the demographic attribute that significantly affects the public's satisfaction with the residential area [46].

## 1.4 Research basis

The front garden of multi-storey residential buildings is composed of various landscape features.

Green area is positively influential to the public [47]. Cerina, Fornara [48] held that residential landscape is a public landscape and greater green area would play a role in soothing pressure and delighting people's mood. Kondo, Fluehr [49] discovered that urban green space is inversely correlated with mortality, human heart rate, and instances of violence.

The type of plants exerts a key impact on the public's preference for landscapes [50, 51]. Landscape plants are common in gardens. However, CoDyre, Fraser [52] revealed that there are also a lot of people who would like to plant crops in their gardens. Kim and Ohara [53] conducted a research in Japan and found that the owners in the first floor prefer planting vegetables in their private gardens. Ambrose, Das [54] also discovered that planting vegetables could produce stronger happiness than planting ornamental plants.

The arrangement of plant landscapes in terms of their vertical structure influences people's preference for garden landscapes [55]. Zhao, Xu [56] discovered that people have a preference for plant landscapes characterized by a diverse vertical structure. Such plant landscapes, featuring multiple vertical layers, align more closely with the visual preferences of residents, consequently holding greater ecological significance.

The diversity of plants is also of great help to enhance the attraction of landscapes. Häfner, Zasada [30] observed that diversified landscapes are the most popular in landscape assessment. The number of colors would influence people's landscape preferences [57]. However, the number of colors should be usually within a range for people's preference. If there are too many colors, the overall structure of the landscape will be destroyed and accordingly the landscape appears messy; if there are too few colors, the landscape appears monotonous and simple [58].

High coverage of vegetation may produce positive landscape preferences [59–61]. Wang, Zhao [50] conducted experiments on visual preference consensus of different landscape types

and found that environments with higher vegetation coverage significantly increases aesthetic preference consensus. Ambrose, Das [54] claimed that as vegetation coverage increases, people's satisfaction with their garden and the amount of time spent in it would increase accordingly.

Fence materials do affect people's visual preference for garden landscapes. Voicu and Been [62] conducted a study of community gardens in the Bronx, New York, USA, which revealed that the attractiveness of fences significantly affect public acceptability of gardens.

Based on literature research, this study classified the landscape features of the front garden of multi-storey residential buildings into six types: size of garden area, plant type, plant vertical structure, color number, plant coverage, and fence materials. The approach of synthesizing and examining target attributes through literature review is a common practice in analogous research papers [63].

### 1.5 Research questions

Through the photos shared by the owners on social media, this study first collected the visual impact assessment data on the front gardens of different multi-storey residential buildings in Chinese cities. Then through large-scale public participation, this study collected the data on the visual impact assessment of front gardens of multi-storey residential buildings rendered by the public with different demographic attributes. On this basis, through statistical analysis, the following questions were to be explored in this study;

(1) Is there any difference between the visual impact assessment of the front garden of multi-storey residential buildings rendered by the owners and that by the public? If yes, what landscape features cause this difference?

(2) Will the public of different demographic attributes render different visual impact assessments of the front garden of multi-storey residential buildings? If yes, what landscape features cause this difference?

## 2. Research method

### 2.1 Research sites

The research sites of this study are located in 13 cities in Jiangsu Province, China, including Suzhou, Wuxi, Changzhou, Nanjing, Nantong, Yangzhou, Yancheng, Taizhou, Zhenjiang, Huai'an, Suqian, Xuzhou, and Lianyungang. These cities are distributed in the southeast coastal area of China with rapid urbanization development. Multi-storey residential buildings have become one of the core components of urban residential construction. In this sense, the research on the front garden of multi-storey residential buildings becomes more representative. This study conducted photo content retrieval on five Chinese social media platforms, including Weibo, Douyin, Zhihu, Xiaohongshu, and Baidu Tieba, using the keyword "front gardens of multi-storey residential buildings." The cutoff date for relevant content retrieval was set at December 2021. During the manual screening process, photos with low quality and those unrelated to front gardens of multi-storey residential buildings were excluded. A total of 853 photos were posted by the owners of front gardens of multi-storey residential buildings on social media platforms. They expressed their satisfaction with their gardens on these social platforms, and then displayed and shared them online. Among these 853 photos, 586 covered only parts of the garden and 267 the entire garden.

### 2.2 Landscape features

Based on previous studies, this study analyzed different front gardens of multi-storey residential buildings, and finally selected six landscape features as the main objects of study. These six

**Table 1. The landscape features of front garden of multi-storey residential buildings.**

| Landscape features | Property |
|---|---|
| Size of garden | small = 1; moderate = 2; large = 3 |
| Plant type | Lawn only = 1; edible plant + lawn = 2; landscape plant + lawn = 3; lawn+ edible plant + landscape plant = 4 |
| Vertical structure | low = 1; moderate = 2; high = 3 |
| Color number | low = 1; moderate = 2; large = 3 |
| Plant coverage | low = 1; moderate = 2; high = 3 |
| Fence material | Hedge fence = 1; wooden fence = 2; mental fence = 3 |

landscape features are: size of garden, plant type, plant vertical structure, color, number, plant coverage, and fence material (Table 1).

(1) Size of garden (denoted as A): In order to better study the impact of the front garden size of the multi-storey residential building on the public's visual impact assessment, the garden area of each photo was estimated based on surrounding references. According to the size of the garden area, the gardens were finally divided into three groups: large (area greater than or equal to 40m$^2$), medium (area greater than or equal to 20m$^2$), and small (area less than or equal to 20m$^2$).

(2) Plant type (denoted as T): This study counted the plant types of all the photos used as experimental materials, and found that lawn existed in almost every front garden of the multi-storey residential buildings. In the end, the gardens were divided into four groups in terms of plant types, namely, lawn only, edible plant + lawn, landscape plant + lawn, and edible plant + landscape plant + lawn.

(3) Plant vertical structure (denoted as H): This study counted the plant vertical structures of all the photos used as experimental materials, and divided the front gardens into three groups in terms of plant vertical structure. To be specific, the vertical structures with 1–2 layers of plants were defined as low, those with 3 layers of plants moderate, and those with more than 3 layers high.

(4) Color number (denoted as N): The color number of plants in the front garden of multi-storey residential buildings in the photos was calculated, and the front gardens were finally divided into three groups in terms of color number, namely, low (1–2 plant colors), moderate (3–4 plant colors), and high (5 or more plant colors).

(5) Plant coverage (denoted as C): Of all the photos used in this study, there is no photo without plants. In order to better explore the influence of plant coverage in front gardens on the public's visual impact assessment, the plant coverage of each photo was calculated. Accordingly, the plant coverage was divided into three levels: high, moderate and low. To be specific, the plant coverage equal to or higher than two thirds was defined as high; the plant coverage smaller than two thirds but larger than one third was defined as moderate; and the plant coverage smaller than one third was defined as low.

(6) Fence material (denoted as M): The materials used in the front garden fence of multi-storey residential buildings are relatively limited. Through the investigation of fence materials in all the photos, it is found that there are mainly three material categories, namely, hedge fence, wooden fence, and metal fence.

## 2.3 The owners' preference

The collected 853 photos (including 586 photos with a partial view of the front garden and 267 photos with a full view of the front garden) of the garden in front of multi-storey residential

**Table 2. Order of photo numbers according to the owners' preference.**

| Landscape feature | Order of photo numbers |
|---|---|
| Size of garden (267 photos total) | small (108 photos)>moderate (93 photos) >large (66 photos) |
| Plant type (853 photos total) | landscape plant + lawn (344 photos)>landscape plant +edible plants+ lawn (207 photos) >edible plants + lawn (182 photos) >lawn only (120 photos) |
| Vertical structure (853 photos total) | high (375 photos) >low (279 photos) >moderate (199 photos) |
| Color number (853 photos total) | large (432 photos) > moderate(315 photos)>small (106photos) |
| Plant coverage (267 photos total) | moderate (124 photos) >low (78 photos)>high (65 photos) |
| Fence material (853 photos total) | wooden fence (397 photos) >metal fence (284 photos)>hedge fence (172 photos) |

buildings were screened according to the six landscape features listed above and sorted according to the number of photos (Table 2). Fig 1 displays the photo samples sorted according to landscape features.

In terms of size of garden, the largest number of front gardens shared on social media are small gardens, followed by moderate and large gardens.

In terms of plant types, the largest number of front gardens shared on social media are gardens with landscape plants and lawn, followed by gardens with all the three types of plants. Gardens with lawn only are the fewest.

In terms of plant vertical structure, the largest number of front gardens shared on social media are gardens with a high vertical structure, followed by gardens with a low vertical structure. The gardens with a moderate vertical structure are the fewest.

**Fig 1. Sample photos of front gardens shared by the owners on social media.**

In terms of plant color number, the owners like to share their colorful gardens on social media. The most-shared gardens are those with a large number of colors, followed by moderate and small numbers of colors.

In terms of plant coverage, many owners of front gardens share on social media how they transform the original lawn and pave the ground. The most-shared front gardens are those with a moderate plant coverage, followed by low and high coverages.

In terms of fence material, the most-shared front gardens on social media are those with wooden fence, followed by metal fence and hedge fence.

## 2.4 Photos for the experiment

From the 853 photos of the front garden of the multi-storey residential building, 267 representative photos with a full view of the garden were screened out. Then these selected photos were sent to five architectural experts and five landscape architects via email. They were asked to screen out a total of twenty photos that best represented the six landscape features. The final selection consisted of 9 photos that were unanimously chosen by all experts for further study. However, the photos taken by the owners of the front gardens may vary in brightness and some other aspects; meanwhile, the subjects of the study may be obscured by people or animals and the unified view height can not be guaranteed. Given these problems, the nine selected photos were simplified and then modeled through SketchUp2021, and processed them using lumion11 to finally export the images because the computer-generated virtual environment images can effectively help people make choices in the hypothetical green design [64]. When rendering the model, the perspective selected was a unified one that could see the whole front garden, and the background weather was a unified clear sky. Meanwhile, in order to avoid the influence of different landscape factors of the building background on the research results, the background of the multi-storey buildings was also unified, and finally nine pictures were obtained (Fig 2). The research method of substituting pictures to real landscapes has been proved to be effective and reliable [65].

## 2.5 Demographic attributes of the participants

In this study, the non-owners of the front gardens were selected as the experimental participants, and four demographic attributes were set as variables that may influence visual impact assessment. The four variables are gender, age, education level, and income. To be specific, based on the age composition data of Jiangsu Province, the participants are categorized as three groups: 18–34 years old, 35–59 years old, and 60 years old and above; according to the education level of the population in Jiangsu Province, the participants are categorized as two groups: receiving higher education and receiving no higher education; according to the resident income and expenditure data of Jiangsu province in 2021, the participants are divided into two groups: disposable yearly income<RMB 47,498, and ≥RMB 47,498. The demographic attributes of the participants are shown in Table 3.

## 2.6 The survey of the public's visual impact assessment

The nine prepared images were printed on A4 full-color photo paper at 300dpi and assembled into a book with their sequence randomized. Then these pictures were presented to the participants of the survey to investigate their visual impact assessments. To make it easier for participants to rate the pictures, every two pictures were printed on one A4 sheet, with a total of five A4 sheets. The nine pictures were randomly presented to residents above the age of 18 in residential areas selected at random across the 13 cities in Jiangsu Province. The participants were then asked to rate the pictures. The survey time was scheduled on weekends to avoid the

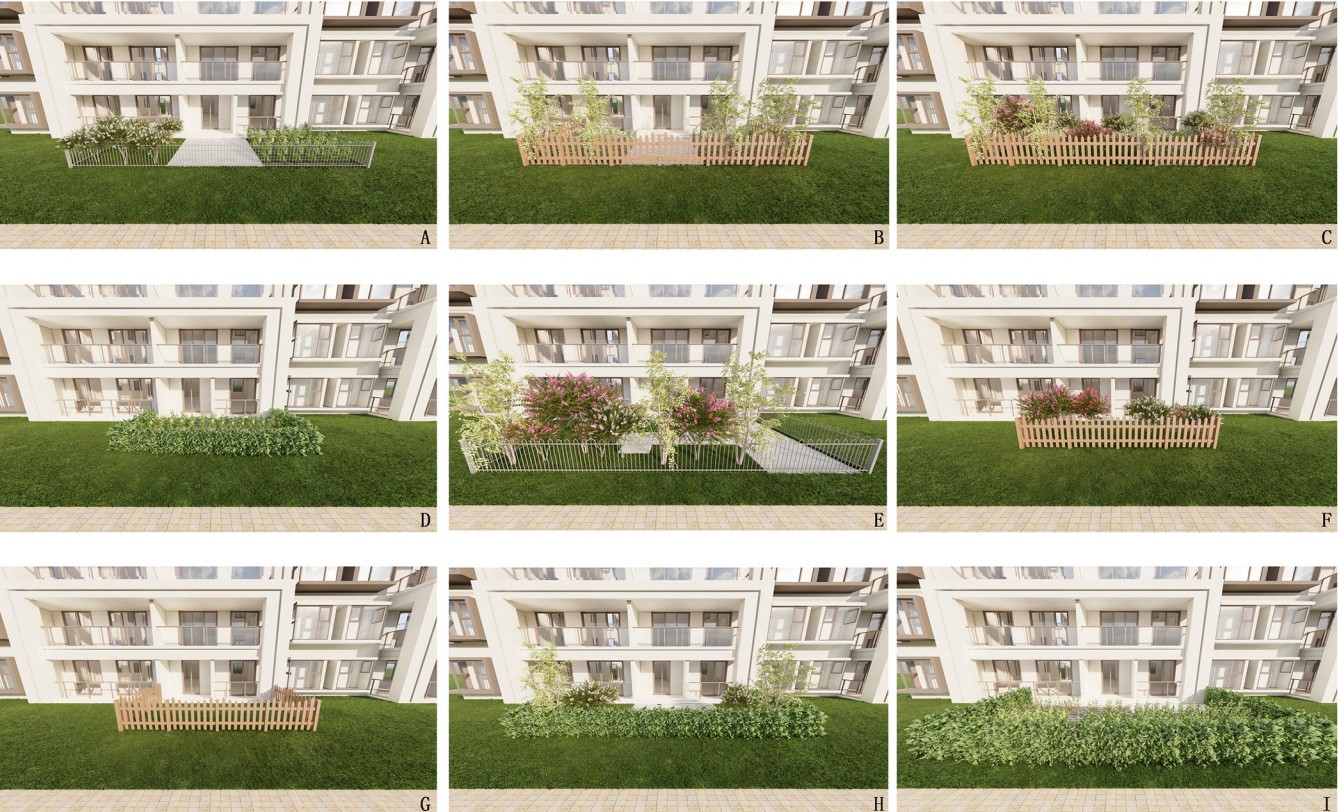

**Fig 2. Pictures for the experiment and their number.**

singularity of demographic variables. Participants were requested to provide information about their demographic attributes, which in this case encompassed gender, age, educational level, and income. After that, they would browse and rate the printed pictures.

The participants were asked to assign a score to each of the nine pictures, within the score range of 0–5, with 0 denoting "strongly dislike" and 5 "strongly like". Throughout this procedure, participants could freely review and modify their scores for any picture at their discretion before finalizing their submissions. No specific permits were required for these activities, because these studies did not involve protected plant species or area of land. All these surveys

**Table 3. Demographic classification of the participants.**

| Demographic attributes | Variable | Set value |
|---|---|---|
| **Gender** | male | 1 |
| | female | 2 |
| **Age** | 18–34 | 1 |
| | 35–59 | 2 |
| | ≥60 | 3 |
| **Education level** | with higher education | 1 |
| | without higher education | 2 |
| **Disposable income/year (Converted at the exchange rate between RMB and USD in 2021)** | <$7,362 | 1 |
| | ≥$7,362 | 2 |

**Table 4. Demographic attributes of the participants.**

| Demographic attributes | Variable | Number of participants | Percentage of participants | Overall proportion of Jiangsu province |
|---|---|---|---|---|
| gender | male | 133 | 51.95% | 50.75% |
| | female | 123 | 48.05% | 49.25% |
| age | 18–34 | 56 | 21.90% | 22.25% |
| | 35–59 | 110 | 42.96% | 38.97% |
| | ≥60 | 90 | 19.78% | 21.84% |
| Education level | with higher education | 57 | 22% | 21.6% |
| | without higher education | 199 | 78% | 78.4% |
| income | <$7,362 | 137 | 53.5% | 53.85% |
| | ≥$7,362 | 119 | 46.5% | 46.15% |

The collected data were analyzed by SPSS 22.0 to explore the influence of different demographic attributes on the visual impact assessment of the six landscape features.

were conducted on the prerequisite of participants' oral consent during this period from February 1, 2022, to March 31, 2022, involving 256 participants in totals, with an effective rate of 88.81%. The demographic attributes of participants were displayed in Table 4. As the statistics shows, the demographic distribution of the participants closely mirrored that of Jiangsu Statistical Yearbook 2022, indicating that the survey was representative.

### 2.7 Ethics statement

The study is not ethically sensitive and was carried out in accordance with national and institutional legal and ethical requirements. The data were collected completely anonymously (i.e. no possibility to reidentify whatsoever) and therefore this work falls outside the scope of GDPR 2016 and MDSM (Measures for Data Security Management) for China.

The study is in line with institutional guidelines; and the internal ethics reference person was consulted, who indicated that there was no need for ethical approval since surveys were not directly health-related. In China, there is no legal requirement for ethical approval for such a survey as long as no sensitive issues or privacy is involved, and there are no IRB mechanisms in place for this type of work. Sensitive data or research involving human subject undergo ethical approval through ethical research committees.

To be specific, ethical concerns were assessed internally: participation was on a voluntary basis and all participants were informed that the survey was anonymous; all data would be only used for research, evaluated anonymously, and not be made public in any form. To secure privacy, all data were collected and analyzed anonymously with no collection of identifiers/codes.

## 3. Results

### 3.1 The overall assessment of the selected pictures

The nine pictures (Fig 2) underwent intergroup reliability testing using SPSS 22.0, yielding a result of 0.757, which signifies a noteworthy degree of internal consistency. Consequently, it can be inferred that the reliability of questionnaire surveys is substantiated, and the collected dataset can be competently used for further detailed analysis.

Each picture's average score, as provided by the participants, is represented by the variable "S". Among these nine pictures, the highest average score is 3.95 while the lowest is 2.37(the scoring range: 0–5). The overall average score for all the pictures is 3.35. Picture F attained the highest average score, whereas picture A attained the lowest. In experiments substituting

Table 5. Stepwise multiple linear regression analysis results.

| Model | | Unstandardized Coefficients | | Standardized Coefficients | t | Sig. | Collinearity Statistics | |
|---|---|---|---|---|---|---|---|---|
| | | B | Std. Error | Beta | | | Tolerance | VIF |
| 1 | (Constant) | 1.274 | .148 | | 8.586 | .013 | | |
| | A | 1.101 | .063 | 2.138 | 17.382 | .003 | .103 | 9.700 |
| | T | -.111 | .028 | -.272 | -3.922 | .049 | .325 | 3.076 |
| | H | .403 | .045 | .639 | 9.021 | .012 | .311 | 3.212 |
| | N | -.518 | .045 | -.822 | -11.600 | .007 | .311 | 3.216 |
| | C | .321 | .036 | .668 | 8.892 | .012 | .276 | 3.621 |
| | M | -.407 | .060 | -.608 | -6.808 | .021 | .196 | 5.109 |

a. Dependent Variable: S2

pictures for actual landscapes, the average scores assigned to the pictures can be considered valid data for evaluating participants' assessments of visual impact [66].

## 3.2 The correlation between landscape features and the public's visual impact assessment

To study the correlation between the six landscape features and visual impact assessment, the initial step involved performing a stepwise multiple linear regression analysis. The size of garden (A), plant type (T), vertical structure of plants (H), color number (N), plant coverage (C), and fence material (M) are taken as constants and the dependent variable chosen for analysis is the average score (S) assigned to each picture. The outcomes of the analysis are presented in Table 5.

As indicated in Table 5, all the six constants (A, T, H, N, C and M) have a substantial impact on the average scores of the pictures. To be specific, A(F = 17.382, p = 0.003); T(F = -3.922, p = 0.049); H(F = 9.021, P = 0.012); N(F = -11.600, P = 0.007); C(F = 8.892, P = 0. 012); M(F = -6.808, P = 0.021).

Therefore, from the results, it could be concluded that when the average score of the pictures S is taken as the dependent variable, there is a notable difference in all the six landscape features of front gardens of multi-storey residential buildings. In other words, all six landscape features affect the average score of the pictures.

## 3.3 The correlation between demographic attributes and the public's visual impact assessment

The relationship between demographic attributes and visual impact assessment was explored using one-way analysis of variance.

As evident from the analysis results, there are notable disparities in the average scores provided by participants based on the following demographic attributes: gender difference (F = 7.324, p = 0.12), age difference (F = 4.342, p = 0.001), educational levels difference (F = 6.353, p = 0.005), and income levels difference (F = 3.555, p = 0.005).

Kendall rank correlation analysis was employed to reassess the relationship between demographic attributes and visual impact assessment. As the analysis results show in Table 6, the average score (S) exhibits a positive correlation with gender, age, educational level, while demonstrating a negative correlation with income.

Stepwise multiple linear regression was conducted for further analysis. In this model, gender, age, education level, and income are taken as independent variables; and the average score

**Table 6. Kendall rank correlation analysis results.**

| | | Gender | Age | Education | Income | Score |
|---|---|---|---|---|---|---|
| Gender | Correlation Coefficient | 1.000 | .113 | .070 | -.083 | .261* |
| | Sig. (2-tailed) | . | .430 | .644 | .582 | .039 |
| | N | 256 | 256 | 256 | 256 | 256 |
| Age | Correlation Coefficient | .113 | 1.000 | .126 | -.258 | .470** |
| | Sig. (2-tailed) | .430 | . | .377 | .071 | .000 |
| | N | 256 | 256 | 256 | 256 | 256 |
| Education | Correlation Coefficient | .070 | .126 | 1.000 | .017 | .253* |
| | Sig. (2-tailed) | .644 | .377 | . | .913 | .046 |
| | N | 256 | 256 | 256 | 256 | 256 |
| Income | Correlation Coefficient | -.083 | -.258 | .017 | 1.000 | -.401** |
| | Sig. (2-tailed) | .582 | .071 | .913 | . | .002 |
| | N | 256 | 256 | 256 | 256 | 256 |
| Score | Correlation Coefficient | .261* | .470** | .253* | -.401** | 1.000 |
| | Sig. (2-tailed) | .039 | .000 | .046 | .002 | . |
| | N | 256 | 256 | 256 | 256 | 256 |

**. Correlation significant at the 0.01 level (2-tailed)

*. Correlation significant at the 0.05 level (2-tailed).

of pictures (S) is set as dependent variable (Table 7). As is indicated by the results, each of the four independent variables significantly contributes to the average scores of the pictures.

In order to study whether there is reciprocal effect among demographic attributes, co-linear analysis of independent variables is performed in accordance with the findings of the multiple linear regression model.

The tolerance of gender is 0.749, VIF = 1.334; that of age is 0.600, VIF = 1.668; that of education level is 0.953, VIF = 1.050; the tolerance of income is 0.711, VIF = 1.406. When the VIF of the model is greater than 10 or the tolerance is smaller than 0.2, the model has collinearity problems [67]. The VIF of the independent variables calculated by SPSS are all smaller than 10 and their tolerances are all greater than 0.2; and the residuals are normally distributed. Therefore, it could be concluded that the model had no collinearity problem.

## 3.4 Demographic differences among participants and landscape features of pictures

The average score of each picture provided by the participants with different demographic attributes was set as a dependent variable; and the six landscape features (A, T, H, N, C and M)

**Table 7. Stepwise multiple linear regression analysis results.**

| Model | | Unstandardized Coefficients | | Standardized Coefficients | t | Sig. | Collinearity Statistics | |
|---|---|---|---|---|---|---|---|---|
| | | B | Std. Error | Beta | | | Tolerance | VIF |
| 1 | (Constant) | 1.713 | .355 | | 4.832 | .000 | | |
| | Gender | .365 | .116 | .330 | 3.154 | .003 | .749 | 1.334 |
| | Age | .255 | .089 | .335 | 2.867 | .007 | .600 | 1.668 |
| | Education | .232 | .103 | .209 | 2.258 | .029 | .953 | 1.050 |
| | Income | -.342 | .123 | -.299 | -2.788 | .008 | .711 | 1.406 |

a. Dependent Variable: Score

**Table 8. Results of stepwise multiple linear regression analysis of the photos-based landscape factors according to different demographic attributes.**

| Dependent | | Unstandardized Coefficients | | Standardized Coefficients | t | Sig. | Collinearity Statistics | |
|---|---|---|---|---|---|---|---|---|
| | | B | Std. Error | Beta | | | Tolerance | VIF |
| Scores for Male | (Constant) | 1.517 | .196 | | 7.744 | .016 | | |
| | A | -.128 | .050 | -.227 | -2.573 | .024 | .532 | 1.879 |
| | C | .622 | .060 | .798 | 10.384 | .009 | .702 | 1.424 |
| | M | -.232 | .046 | -.460 | -5.062 | .037 | .502 | 1.990 |
| Scores for Female | (Constant) | .210 | .304 | | .691 | .561 | | |
| | T | -.348 | .032 | -.630 | -11.042 | .008 | .457 | 2.190 |
| | H | .470 | .043 | .457 | 10.855 | .008 | .837 | 1.195 |
| | N | .193 | .055 | .261 | 3.486 | .043 | .265 | 3.771 |
| 18–35 Years of Age | (Constant) | 2.504 | .381 | | 6.569 | .022 | | |
| | T | -.489 | .068 | -1.015 | -7.200 | .019 | .383 | 2.609 |
| | N | .572 | .109 | .752 | 5.237 | .035 | .370 | 2.706 |
| | M | -.495 | .100 | -.719 | -4.955 | .038 | .361 | 2.768 |
| 36–59 Years of Age | (Constant) | .853 | .236 | | 3.616 | .069 | | |
| | A | .966 | .086 | 2.244 | 11.264 | .008 | .139 | 7.216 |
| | N | -.280 | .063 | -.627 | -4.437 | .047 | .275 | 3.631 |
| | C | .535 | .082 | 1.333 | 6.561 | .022 | .133 | 7.505 |
| 60 Years of Age or Older | (Constant) | 2.555 | .153 | | 16.662 | .004 | | |
| | T | -.102 | .021 | -.210 | -4.771 | .041 | .316 | 3.166 |
| | H | .416 | .055 | .487 | 7.566 | .017 | .148 | 6.760 |
| | C | -.149 | .020 | -.270 | -7.640 | .015 | .492 | 2.034 |
| | M | -.190 | .025 | -.414 | -7.629 | .017 | .207 | 4.820 |
| Higher education | (Constant) | 2.020 | .179 | | 11.290 | .008 | | |
| | T | -.441 | .030 | -1.082 | -14.924 | .004 | .187 | 5.355 |
| | H | .437 | .063 | .677 | 6.891 | .020 | .102 | 9.846 |
| | N | .689 | .059 | 1.068 | 11.632 | .007 | .116 | 8.586 |
| | M | .223 | .050 | .382 | 4.425 | .047 | .132 | 7.597 |
| Without higher education | (Constant) | 1.254 | .127 | | 9.879 | .010 | | |
| | T | -.488 | .026 | -1.088 | -18.590 | .003 | .377 | 2.652 |
| | C | .352 | .025 | .585 | 14.309 | .005 | .774 | 1.293 |
| | M | .195 | .029 | .324 | 6.616 | .022 | .538 | 1.860 |
| <$7,362 | (Constant) | 3.285 | .086 | | 38.124 | .001 | | |
| | A | -.191 | .038 | -.622 | -5.034 | .037 | .146 | 6.838 |
| | T | .149 | .018 | .626 | 8.077 | .015 | .372 | 2.691 |
| | H | -.365 | .047 | -.973 | -7.759 | .016 | .142 | 7.051 |
| ≥$7,362 | (Constant) | 1.283 | .151 | | 8.496 | .014 | | |
| | T | .370 | .019 | .861 | 19.621 | .003 | .702 | 1.424 |
| | H | -.184 | .062 | -.308 | -2.974 | .027 | .126 | 7.935 |
| | N | -.246 | .044 | -.373 | -5.622 | .030 | .308 | 3.251 |
| | C | .545 | .049 | .881 | 11.088 | .008 | .214 | 4.666 |

are set as independent variables. As is indicated by the stepwise multiple linear regression models, gender, age, educational level, and income each yielded distinct significant predictors (Table 8).

For male participants, A, C and M are reliable predictors. They prefer the front gardens with moderate size, high plant coverage, and hedge fence. Regarding female participants, T, H,

and N are reliable predictors, and they prefer the front gardens with landscape plants, high vertical structure, and 3–4 colors.

For the participants of 18–34 years old, T, N and M are reliable predictors; they prefer the front gardens with landscape plants +lawn, moderate color number, and wooden fence. Regarding the participants of 35–59 years old, A, N and C are reliable predictors; they prefer the front gardens with moderate size, high plant coverage, and 3–4 colors. Regarding the participants of 60 years old and above, T, H, C and M are reliable predictors for visual impact assessment; they prefer the front gardens with edible plants +lawn, moderate vertical structure, high plant coverage, and wooden fence.

For the participants who have attained higher education, T, H, N and M are reliable predictors; they prefer the front gardens with landscape plants, moderate vertical structure, 3–4 colors, and hedge fence. For the participants who do not receive higher education, T, C, and M are reliable predictors; they prefer the front gardens with edible plants + lawn, high plant coverage, and wooden fence.

For participants whose income is below $7,362, A, T and H are reliable predictors; they prefer the front gardens with small size, moderate vertical structure, and edible plants + lawn. For participants whose income reaches or exceeds $7,362, T, H, N, and C are reliable predictors; they prefer the front gardens with landscape plants + lawn, high vertical structure, high plant coverage and 1–2 colors.

## 4. Discussions

### 4.1 Difference between owners and the public in their visual impact assessments

There exists significant difference between the visual impact assessment of the front garden of multi-storey residential buildings rendered by the owners and that by the public.

**(1) Size of garden.** The size of front garden affects the visual impact assessment. As is demonstrated by this study, the front garden whose area is over 40m$^2$ is scored the lowest; while the front garden with a moderate size is scored the highest. This may be because the front garden of multi-storey residential buildings is essentially part of the greening of the community, and a large garden area may be perceived as the owner transforming too much public space into a private property, causing public dissatisfaction [68].

**(2) Plant type.** Plant type also affects the public' visual impact assessment. The combination of landscape plants +lawn boasts of the greatest popularity among the public, which is in line with the front garden owners' preference (The preferences of front garden owners of multi-storey residential buildings are derived from Table 2). This is possibly because the public pay more attention to its landscape beauty when observing the front garden of the multi-storey buildings, and the landscape plants can bring them certain visual enjoyment. For the owners of the front garden, the combination of all the three plant types and that of edible plants +lawn are more favorable than lawn only. According to Lindemann-Matthies and Brieger [69], urban green spaces exhibit greater aesthetic appeal when incorporating flower beds or fields and well-organized vegetable plots, as opposed to traditional lawns. Contrary to their conclusion, the front garden with a lawn only is scored slightly higher than those with all three plant types or edible plants +lawn. This may be due to the fact that the public prefer to see simple and accessible front gardens, while gardens with both landscape and edible plants appear cluttered and lack public character.

**(3) Vertical structure of plants.** The vertical structure of plants in the front garden of multi-storey buildings also affect the public's visual impact assessment. To be more precise, the front garden picture with low vertical structure is scored relatively lower, which is

consistent with the owners' preference. This may be because the public prefer landscapes with multiple layers, which is similar to the findings obtained by Zhao [70]. However, different from the owners of front gardens, the public prefer the gardens with moderate vertical structure rather than high vertical structure. This may be due to the relatively high density of plants in a garden with a high vertical structure, which makes the garden appear more enclosed.

**(4) Plant color number.**   The owners of front gardens prefer colorful gardens. To be specific, they prefer the gardens with a large number of colors, followed by those with a moderate number of colors. The garden with a small number of colors is the least popular among the owners. Huang, Han [71] observed that the color number of plants is positively correlated with the public's preference. However, contrary to their findings and the owners' preference, the public score the garden with 3–4 colors of plants the highest, while the lowest score goes to the gardens with five or even more colors of plants. This may be because too many colors would mess up the overall appearance of the garden. In contrast, plant landscapes with only 1–2 colors are scored slightly higher than those with five or more colors. This may be attributed to the fact that plant landscapes with only 1–2 colors appear too monotonous but still relatively more attractive than those with five or more colors. Accordingly, the gardens with 3–4 plant colors align better with the aesthetic preferences of the public.

**(5) Plant coverage.**   Plant coverage also exerts some certain influence on the public's visual impact assessment. Diametrically different from the owners' preference, the front gardens with a high plant coverage is scored the highest while the garden with a low plant coverage is scored the lowest by the public. This may be because the public generally think that the owner of the front garden only have the right to use the garden and they prefer to see more natural elements rather than large areas of hard paving exclusively for the first floor owners. Ambrose, Das [54] also found that as the plant coverage increases, the public would spend more time in the garden and their satisfaction increases accordingly.

**(6) Fence material.**   The materials used for the fence of front gardens of multi-storey residential buildings also affect visual impact assessment. The owners of front gardens prefer the wooden fence, followed by metal fence and hedge fence. To the public, the wooden fence is scored the highest, followed by hedge fence and metal fence. This may be because wooden fence is the most commonly seen in the gardens or yards in the countryside in China [72], thus being highly familiar and friendly. Compared with metal fence which is cold and impersonal, wooden fence embodies more natural elements in the garden; hedge fence may block the public's view due to its low permeability.

## 4.2 Demographic attributes and the public's visual impact assessment

The public of different demographic attributes rendered different visual preference assessments of the front garden of multi-storey residential buildings.

**(1) Gender.**   Gender difference may result in different visual impact assessments of front gardens. According to the statistical results obtained, female participants would score the front gardens higher than male ones. This corroborates prior research findings that female are more positive toward environment and nature than males [73].

**(2) Age.**   As people grow older, they become more and more concerned about and sensitive to the social and natural environment [74]. The age of the participants exhibits a positive correlation with the average scores of the pictures in this study. Stated differently, the older the participants become, the higher they would score the pictures. This may be related with their living environment, growth experience and other factors [36]. Most likely, the senior participants move to cities from the countryside to take their children or even grandchildren, or simply to be taken care of by their children since they are too old to live alone in the countryside.

Due to their experience of living in the countryside for a long time, they have special attachment and affection with gardens and natural environment [75].

(3) **Educational level.** Häfner, Zasada [30] held that the educational level of the participants exerts some significant influence on their visual impact assessment of landscapes, which is consistent with the findings of this study. To be specific, the participants receiving no higher education score the pictures higher than those with higher education. This may be because the participants with higher education may have some knowledge about gardening and landscapes and have appreciated many landscapes. Accordingly, they would be stricter with their visual impact assessment of front gardens and score the pictures relatively lower. In contrast, the participants without any higher education may know little about landscape and gardening and their assessment would be more subjective and intuitive. As a result, they may produce higher assessments of front gardens more easily.

(4) **Income.** Waddick [41] maintained that the public with different incomes would produce different visual impact assessments, which corresponds with the conclusions drawn from this study. As is shown by the statistical results, the average scores of pictures given by the participants whose yearly disposal income reaches or exceeds $7,362 are lower than those given by the participants whose yearly disposal income is lower than $7,362. This may be because the public with higher income are more inclined to maintain the integrity of community greening. They may prefer the community greening which has not been altered by the owners of first-floor residence privately. To them, the existence of front gardens may be considered a factor that disrupts the integrity of the community greening.

The available evidence regarding the impact of gender, age, income, and education on visual preference is not yet definitive. Hence, additional research is warranted to investigate how demographic attributes affect the evaluation of landscape visual impact.

## 4.3 Demographic attributes and landscape features

(1) **Gender and landscape features.** Male participants would give their priority to the size of garden, plant coverage and fence material when they assess the front gardens of multi-storey residential buildings. This may be because men are more inclined to holistic thinking, and place greater emphasis on the overall image of the garden. Meanwhile, men prefer the front gardens with high plant coverage and fence made of natural materials. When assessing the front gardens of multi-storey residential buildings, women attach more importance to plant type, vertical structure, and color number. Plant type is the most influential factor for their assessment. They usually prefer the gardens with landscape plants, which may be due to the fact that women would inherently focus on beautiful things in terms of aesthetics. In addition, they pay more attention to details. Accordingly, they display an increasing preference for landscapes with rich and diversified elements [30].

(2) **Age and landscape features.** As is revealed in this experiment, the participants of 18–34 years old place greater focus on plant type, color number and fence material. This may be because people of this age group mostly grow up in cities and thus are more sensitive and acceptant to newly-emerging things. Therefore, they prefer the front gardens with landscape plants +lawn, a moderate number of colors, and wooden fence. The participants who are 35–59 years old attach more importance to the size of garden, color number and plant coverage. People of this age group have accumulated certain wealth and paid more attention to the quality of daily life. Therefore, they prefer the front gardens with a moderate number of colors, high plant coverage, and a moderate area. For the participants who are over 60 years old, the main influencing factor for their visual impact assessment of front gardens is plant type. They display a greater preference for planting edible plants in the garden. This is inconsistent with

the findings of Lindemann-Matthies and Brieger [69]. According to Lindemann-Matthies and Brieger, the elderly prefer neat gardens instead of vegetable patches or loose lawn. In addition, they display a strong preference for the front gardens with moderate vertical structure and high plant coverage. This may be due to their rich life experience in the countryside and abundant knowledge about various edible plants. Compared with hedge fence, wooden fence is featured with better permeability, which enables the elderly to observe the growth of edible plants better and also aligns with their memory of countryside life.

**(3) Educational level and landscape features.** The participants with higher education place their focus more on plant type, vertical structure of plants, color number, and fence material when they assess front gardens of multi-storey buildings. According to the previous research, participants with higher education level display a strong dislike for the lawn where plant species are single and scarce [76]. This is consistent with the findings obtained in this study. The participants with higher education prefer the gardens with landscape plants, which may be because they believe that gardens are essentially community greenery and should have a higher ornamental value than food value. Häfner, Zasada [30] discovered that the participants with higher education exhibit an increasing preference for landscapes with rich and diversified elements. This is also in line with the findings obtained in this study. The participants with higher education prefer the gardens with a moderate vertical structure, 3–4 colors, and hedge fence. This can be explained by the fact that these participants have certain knowledge about landscapes and gardening, thus attaching more importance to the natural attribute of front gardens. In contrast, the participants without higher education prefer the gardens with high plant coverage, edible plants+ lawn, and wooden fence.

**(4) Income and landscape features.** Participants with a yearly disposable income below $7,362 place their priority upon the size of garden, plant type, and vertical structure when they assess front gardens of multi-storey residential buildings. They prefer the gardens with small area, moderate vertical structure, and edible plants + lawn. For the participants with a relatively low income, community greening is a place of free leisure and entertainment [77]. Therefore, they do not want the greening to be much occupied by private owners, nor do they tolerate their eyesight to be blocked by high and dense plants. Meanwhile, they have some certain preference for edible plants. In contrast, participants with a yearly disposable income reaches and exceeds $7,362 place their priority upon plant type, color number, coverage and vertical structure of plants when they assess the front gardens of multi-storey residential buildings. They prefer the gardens with high vertical structure, high plant coverage, 1–2 colors, and a combination of landscape plants and lawn. This can be attributed to the fact that they have already accumulated certain economic power, thus paying more attention to the agreeable living environment.

## 4.4 Limitations

This study explored differences in the public's visual impact assessment of front gardens of multi-storey residential buildings, but did not fully consider the changes of plant image in four seasons, especially in winter. The image of edible plants tends to change greatly with the change of seasons, and how this change will affect the public's visual impact assessment needs to be further studied.

## 5. Conclusions

As a special form of garden, the front garden of multi-storey residential buildings needs to be designed with more consideration for the needs of the public in order to create an environment more in line with their expectations. Housing initiatives catering to the requirements of

substantial demographics are more inclined to garner public backing [78]. As is indicated by the findings of this study, significant differences exist between the owners and the public in their visual impact assessments of front gardens of multi-storey residential buildings. The public wish this special garden could be more accessible and open. In Asian societies, people crave intimacy and contact [79]. The shared and open design of the front gardens can promote the harmony of the neighborhood and enhance the common happiness of the owners and the public. Meanwhile, this study explored the relationship between the landscape features of front gardens of multi-storey residential buildings and visual impact assessment. The results show that all the six landscape features, namely, size of garden, plant type, vertical structure, plant color number, plant coverage, and fence material, exert some certain impact on the public's visual impact assessment.

As is indicated by the results of this study, in order to reach a consensus between the owners and the public, the following factors need to be considered in the design of the front garden of multi-storey residential buildings: the front garden should be moderate in size, with a combination of landscape plants and lawn so as to provide aesthetic perception and visual enjoyment. The design should consider creating a rich vertical structure in the front garden, which can endow the garden with a sense of depth and beauty. Meanwhile too high and dense plants should also be avoided in the periphery so as not to block the view; and the number of plant colors should be controlled within 3–4 colors so that the garden would appear colorful but not messy due to too many colors. Moreover, the hard pavement area should be minimized so that the plant coverage could be sufficient with natural elements and aesthetic perception. Finally, the fence should be wooden so as to create a feeling of naturalness and intimacy.

Admittedly, this study still has some limitations. However, the results obtained from experiments and analysis are of great significance for improving the design of front garden of multi-storey residential buildings. By taking into account the needs of the public and the interests of the owners, it is possible to create a living environment that is more in line with the expectations of the public, thereby reducing potential conflicts and disputes and enhancing the overall quality of the community and the living experience.

## Supporting information

**S1 File.**
(DOCX)

## Acknowledgments

We wish to extend our sincere gratitude to the numerous scholars, experts, and countless anonymous participants.

## Author Contributions

**Conceptualization:** Chenping Han, Mengmeng Zhao, Zhongju Yang, Jian Zhang.

**Formal analysis:** Zhongju Yang.

**Funding acquisition:** Chenping Han, Jian Zhang.

**Investigation:** Lin Zhang.

**Methodology:** Mengmeng Zhao.

**Supervision:** Chenping Han, Jian Zhang.

**Writing – original draft:** Tao Gu, Jian Zhang.

**Writing – review & editing:** Chenping Han, Mengmeng Zhao, Lin Zhang, Jian Zhang.

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
