## [Decision Letter · Decision Letter 0]

31 Oct 2023

PONE-D-23-31026Is there any difference between the owners and the public in their visual impact assessments?——A case study of the front garden of multi-storey residential buildingsPLOS ONE

Dear Dr. Han,

Thank you for submitting your manuscript to PLOS ONE. After careful consideration, we feel that it has merit but does not fully meet PLOS ONE’s publication criteria as it currently stands. Therefore, we invite you to submit a revised version of the manuscript that addresses the points raised during the review process.

We look forward to receiving your revised manuscript.

Kind regards,

Grigorios L. Kyriakopoulos, 2 PhDs, 3 MSc, 2 MA, MEng, 2 BA, BSc

Academic Editor

PLOS ONE

Journal Requirements:

Reviewers' comments:

Reviewer's Responses to Questions

**Comments to the Author**

1. Is the manuscript technically sound, and do the data support the conclusions?

Reviewer #1: Yes

Reviewer #2: Yes

2. Has the statistical analysis been performed appropriately and rigorously? 

Reviewer #1: No

Reviewer #2: Yes

3. Have the authors made all data underlying the findings in their manuscript fully available?

Reviewer #1: Yes

Reviewer #2: Yes

4. Is the manuscript presented in an intelligible fashion and written in standard English?

Reviewer #1: No

Reviewer #2: Yes

5. Review Comments to the Author

Reviewer #1: This manuscript set the size of garden, plant type, plant vertical structure, color number, plant coverage, and fence material as the landscape features to be explored. Then the experiments were conducted to reveal the visual impact assessment rendered by the public with different demographic attributes and the results obtained were analyzed. While the context and analysis were appropriate.

1. The first and biggest issue in this manuscript is the English writing. In general, writing improvements are needed to make this article easier to understand. The manuscript needs to be revised by a professorial English editor to improve the quality and readability, or it would not be able to publish.

2. There are lack of citations in many paragraphs and sections, such as Introduction and Discussions, and so on.

3. Abstract. Some references noted that the readability standard for one sentence is no longer than 20 words. For example, “With the front garden of multi-storey residential buildings in Jiangsu province, China, as research object, this study set the size of garden, plant type, plant vertical structure, color number, plant coverage, and fence material as the landscape features to be explored.” It seems like the second sentence of abstract has over 40 words, it is too long and reduces the readability. and I also suggest checking the entire text.

4. Introduction. Important information in the first, second, and second paragraphs lack supporting references. I suggest revising the phrase "However, there is relatively little research on differences in visual impact assessments between homeowners and the public." It does not clearly reflect the research significance of the article.

5. Research questions. A paper that focuses on solving one or two specific research questions is considered excellent. However, you want to solve four specific research questions in the paper at the same time, scattered research questions may make readers think that you do not think deeply about the problem to be solved in the paper, and may not be able to really solve the problem. It is recommended that you read more literature and refine your research questions.

6.Literature review. It is suggested that you supplement the literature review and find out the theoretical gap of the research question, so as to reflect the research value of your article.

7. Results. “The nine pictures underwent intergroup reliability testing using SPSS 22.0, yielding a result of 0.757, signifying a noteworthy degree of internal consistency. Consequently, it can be inferred that the questionnaire survey's reliability was substantiated, and the collected dataset can be competently used for further detailed analysis.” Please suggest you specify which picture, Figure 2? “For participants whose income is below RMB 47,498”

Please convert it into a monetary quantity in $, so as to facilitate the understanding of researchers and direct communication in a wider range.

8. Discussions. It is recommended that you talk with relevant literature results or related theories. Please respond to what research questions (4 research questions) you have really solved, what is the limitation of the research?

Reviewer #2: Thank you for the opportunity to review your paper “Is there any difference between the owners and the public in their visual impact assessments? A case study of the front garden of multi-storey residential buildings.” as submitted to Journal of PlosONE.

This paper presents an effective and concise approach in a very important subject in the field of y difference between the owners and the public in their visual impact assessments. However, in the introduction is missing the discussion how the resilience of the buildings in the recommended approach can be affected (see below Tampekis et al 2023, Mitoulis et al 2023, Tsantopoulos et al 2018) lines 61-65.

The abstract emphasizes in the scope of the paper and at the same time presents the theoretical framework. The introduction is relevant and covers all the aspects of the recommended theme. Also, the bibliography is updated and theory based.

The results and the discussion are presented well-written.

Finally, the discussion is very helpful and shows the effectiveness of the suggested framework.

6. PLOS authors have the option to publish the peer review history of their article (what does this mean?). If published, this will include your full peer review and any attached files.

Reviewer #1: No

Reviewer #2: No

---

## [Author Response · Author response to Decision Letter 0]

12 Dec 2023

Dear Editors and Reviewers:

Thank you for your letter and for the reviewers’ comments concerning our manuscript entitled “Is there any difference between the owners and the public in their visual impact assessments?——A case study of the front garden of multi-storey residential buildings” (ID: PONE-D-23-31026). Those comments are all valuable and very helpful for revising and improving our paper, as well as the important guiding significance to our researches. We have studied comments carefully and have made correction which we hope meet with approval. Revised portion are marked in red in the “Revised Manuscript with Track Changes”. The main corrections in the paper and the responds to the reviewer’s comments are as flowing:

Responds to the journal requirements:

1. Response to requirement: (“Please ensure that your manuscript meets PLOS ONE's style requirements, including those for file naming.”)

Response: We appreciate your guidance. The article has been adapted to meet PLOS ONE's style requirements, ensuring compliance. Thank you for your feedback.

2. Response to requirement: (“In your Data Availability statement, you have not specified where the minimal data set underlying the results described in your manuscript can be found. PLOS defines a study's minimal data set as the underlying data used to reach the conclusions drawn in the manuscript and any additional data required to replicate the reported study findings in their entirety. All PLOS journals require that the minimal data set be made fully available. For more information about our data policy, please see http://journals.plos.org/plosone/s/data-availability.

Important: If there are ethical or legal restrictions to sharing your data publicly, please explain these restrictions in detail. Please see our guidelines for more information on what we consider unacceptable restrictions to publicly sharing data: http://journals.plos.org/plosone/s/data-availability#loc-unacceptable-data-access-restrictions. Note that it is not acceptable for the authors to be the sole named individuals responsible for ensuring data access.We will update your Data Availability statement to reflect the information you provide in your cover letter.”)

Response: Thank you for your guidance. We have addressed the issue by uploading our study's minimal underlying data set as a Supporting Information file.

Responds to the reviewer’s comments:

Reviewer #1:

1. Response to comment: (“The first and biggest issue in this manuscript is the English writing. In general, writing improvements are needed to make this article easier to understand. The manuscript needs to be revised by a professorial English editor to improve the quality and readability, or it would not be able to publish.”)

Response: We appreciate your guidance, and in response to your feedback, we have taken the necessary steps to address the English writing issues in the manuscript. We have engaged a professional English editor who has conducted a thorough revision to enhance the overall quality and readability. We believe these revisions significantly improve the manuscript, making it more suitable for publication.

2. Response to comment: (“There are lack of citations in many paragraphs and sections, such as Introduction and Discussions, and so on.”)

Response: Thank you for your thorough review and constructive feedback. We have carefully addressed the issue of missing citations in the manuscript. Relevant references have been added at appropriate places in the text, particularly in the Introduction and Discussions sections.

3. Response to comment: (“Abstract. Some references noted that the readability standard for one sentence is no longer than 20 words. For example, “With the front garden of multi-storey residential buildings in Jiangsu province, China, as research object, this study set the size of garden, plant type, plant vertical structure, color number, plant coverage, and fence material as the landscape features to be explored.” It seems like the second sentence of abstract has over 40 words, it is too long and reduces the readability. and I also suggest checking the entire text.”)

Response: Thank you for bringing the issue of readability to our attention. We have revised a number of sentences to try to keep them to no more than 20 words each, wherever possible, in order to comply with the recommended readability standards. In addition, we have extended this revision to other parts of the manuscript to ensure improved readability throughout.

4. Response to comment: (“Introduction. Important information in the first, second, and second paragraphs lack supporting references. I suggest revising the phrase "However, there is relatively little research on differences in visual impact assessments between homeowners and the public." It does not clearly reflect the research significance of the article.”)

Response: Thank you for your feedback. We have carefully reviewed the Introduction section and addressed the issue of missing references for important information in the first, second, and third paragraphs. Relevant supporting references have been added to strengthen the academic foundation of the content. We appreciate your suggestion to revise the phrase, "However, there is relatively little research on differences in visual impact assessments between homeowners and the public." In response to your guidance, we have rephrased it to "However, there is relatively little research on differences in visual impact assessments between homeowners and the public of the front garden of multi-storey residential buildings."We believe this modification more precisely conveys the research significance of our article.

5. Response to comment: (“Research questions. A paper that focuses on solving one or two specific research questions is considered excellent. However, you want to solve four specific research questions in the paper at the same time, scattered research questions may make readers think that you do not think deeply about the problem to be solved in the paper, and may not be able to really solve the problem. It is recommended that you read more literature and refine your research questions.”)

Response: We deeply appreciate the constructive feedback provided by the reviewer. Taking your insights into serious consideration, we have refined our approach by consolidating the four specific research questions into two more focused inquiries. We believe this modification will enable us to delve deeper into the core issues, thereby enhancing the overall quality and coherence of our research.

6. Response to comment: (“Literature review. It is suggested that you supplement the literature review and find out the theoretical gap of the research question, so as to reflect the research value of your article.”)

Response: Thank you for your suggestion. We have made additions to the literature review. We believe these enhancements strengthen the overall research value of our article. Your feedback has been instrumental in guiding these improvements, and we are grateful for your valuable input.

7. Response to comment: (“Results. ‘The nine pictures underwent intergroup reliability testing using SPSS 22.0, yielding a result of 0.757, signifying a noteworthy degree of internal consistency. Consequently, it can be inferred that the questionnaire survey's reliability was substantiated, and the collected dataset can be competently used for further detailed analysis.’ Please suggest you specify which picture, Figure 2? ‘For participants whose income is below RMB 47,498’. Please convert it into a monetary quantity in $, so as to facilitate the understanding of researchers and direct communication in a wider range.”)

Response: We are very sorry for our negligence. We apologize for not making it clear here that the images for which intergroup reliability testing was conducted in this sentence refer to the nine images in Figure 2. We have now labeled them in the text. 

To facilitate broader comprehension and seamless communication, we have converted the mentioned income threshold "below RMB 47,498" into its equivalent in US dollars.

8. Response to comment: (“Discussions. It is recommended that you talk with relevant literature results or related theories. Please respond to what research questions (4 research questions) you have really solved, what is the limitation of the research?”)

Response: Thank you for your valuable feedback on the Research Questions and Discussions sections. We sincerely appreciate your guidance throughout this process.

In response to your recommendations, we have provided detailed responses to the two research questions outlined in the Introduction section, aiming to thoroughly address the core aspects of our study. Specifically, in the opening section of Discussions 4.1 and 4.2, we have articulated that there is a significant difference between the visual impact assessments of the front garden of multi-storey residential buildings as perceived by owners and the public. Furthermore, we have highlighted that the public, with different demographic attributes, demonstrates varying visual preference assessments, and we've elaborated on how specific landscape elements influence these assessments.

Additionally, we have taken into consideration the limitations of our article, explicitly outlining them in the statement: " This study explored differences in the public’s visual impact assessment of front gardens of multi-storey residential buildings, but did not fully consider the changes of plant image in four seasons, especially in winter. The image of edible plants tends to change greatly with the change of seasons, and how this change will affect the public’s visual impact assessment needs to be further studied."

We are grateful for your thoughtful review and suggestions. If you have any further comments or specific areas that require attention, please feel free to let us know. Your feedback has been instrumental in refining the precision and coherence of our manuscript.

Reviewer #2: 

This paper presents an effective and concise approach in a very important subject in the field of y difference between the owners and the public in their visual impact assessments. However, in the introduction is missing the discussion how the resilience of the buildings in the recommended approach can be affected (see below Tampekis et al 2023, Mitoulis et al 2023, Tsantopoulos et al 2018) lines 61-65.

Response: We appreciate your positive assessment of our paper. We acknowledge the gap you've identified in the introduction regarding the discussion of how the recommended approach may impact the resilience of the buildings. Thank you for your valuable feedback. We have addressed the gap in the introduction, and a discussion on building resilience has been added to the presentation. We appreciate your insights.

Special thanks to you for your good comments.

We tried our best to improve the manuscript and made some changes in the manuscript. These changes will not influence the content and framework of the paper. We appreciate for editors and reviewers’ warm work earnestly, and hope that the correction will meet with approval.

Once again, thank you very much for your comments and suggestions.

Thank you and best regards.

Yours sincerely,

Corresponding author:

Name: Chenping Han

E-mail: hanchengping@cumt.edu.cn

---

## [Editor Report · Decision Letter 1]

14 Dec 2023

Is there any difference between the owners and the public in their visual impact assessments?——A case study of the front garden of multi-storey residential buildings

PONE-D-23-31026R1

Dear Dr. Han,

We’re pleased to inform you that your manuscript has been judged scientifically suitable for publication and will be formally accepted for publication once it meets all outstanding technical requirements.

Kind regards,

Grigorios L. Kyriakopoulos, 2 PhDs, 3 MSc, 2 MA, MEng, 2 BA, BSc

Academic Editor

PLOS ONE
---

## [Editor Report · Acceptance letter]

22 Dec 2023

PONE-D-23-31026R1 

PLOS ONE

Dear Dr. Han, 

I'm pleased to inform you that your manuscript has been deemed suitable for publication in PLOS ONE. Congratulations! Your manuscript is now being handed over to our production team.

Kind regards, 

on behalf of

Dr. Grigorios L. Kyriakopoulos 

Academic Editor

PLOS ONE